# Mechanistic and structural basis for activation of cardiac myosin force production by omecamtiv mecarbil

Vicente J. Planelles-Herrero [1,2], James J. Hartman [3], Julien Robert-Paganin[1], Fady I. Malik[3] & Anne Houdusse [1]

Omecamtiv mecarbil is a selective, small-molecule activator of cardiac myosin that is being developed as a potential treatment for heart failure with reduced ejection fraction. Here we determine the crystal structure of cardiac myosin in the pre-powerstroke state, the most relevant state suggested by kinetic studies, both with (2.45 Å) and without (3.10 Å) omecamtiv mecarbil bound. Omecamtiv mecarbil does not change the motor mechanism nor does it influence myosin structure. Instead, omecamtiv mecarbil binds to an allosteric site that stabilizes the lever arm in a primed position resulting in accumulation of cardiac myosin in the primed state prior to onset of cardiac contraction, thus increasing the number of heads that can bind to the actin filament and undergo a powerstroke once the cardiac cycle starts. The mechanism of action of omecamtiv mecarbil also provides insights into uncovering how force is generated by molecular motors.

[1] Structural Motility, Institut Curie, PSL Research University, CNRS, UMR 144, F-75005 Paris, France. [2] Sorbonne Universités, UPMC Univ Paris06, Sorbonne Universités, IFD, 4 Place Jussieu, 75252 Paris cedex 05, France. [3] Research and Development, Cytokinetics, Inc., South San Francisco, CA 94080, USA. Correspondence and requests for materials should be addressed to A.H. (email: anne.houdusse@curie.fr)

Heart failure is a common human condition and is the most frequent cause of hospitalization in people over the age of 65[1]. Once hospitalized for heart failure, mortality rates at 30 days, 1 year, and 5 years were as high as 10%, 22%, and 42% in patients from four United States communities[2]. Decreased cardiac contractility is a central feature in systolic heart failure also known as heart failure with reduced ejection fraction (HF-rEF), and yet there exist no medications that directly enhance cardiac contractility and simultaneously improve survival.

Targeting cardiac myosin may be a promising approach for the treatment of systolic heart failure, increasing cardiac contractility in the absence of deleterious adverse effects such as tachycardia, hypotension, and cardiac arrhythmias[3]. Omecamtiv mecarbil (OM) is a selective, small-molecule cardiac myosin activator that binds to the catalytic domain of myosin and increases cardiac contractility in preclinical models without affecting cardiac myocyte intracellular calcium concentrations or myocardial oxygen consumption[4, 5]. The effects of OM on cardiac function have been studied extensively in humans[3, 6, 7]. Currently, OM is being studied in a Phase 3 clinical trial of ~8000 patients to determine if treatment with OM, when added to standard of care, reduces the risk of cardiovascular death or heart failure events in patients with chronic heart failure and reduced ejection fraction (GALACTIC-HF, www.clinicaltrials.gov identifier, NCT02929329).

Several studies have characterized how OM influences the transitions of the motor ATPase cycle[4, 8, 9]. However, the structural basis that reveals how it enhances force generation is currently unknown. Understanding its precise mechanism of action could shed light on critical questions regarding force generation by the myosin motor and the function of the sarcomere that are still a matter of debate.

The force produced by myosin motors is generated by coupling the sequential release of the products of ATP hydrolysis (Phosphate ($P_i$) and ADP) to a series of strong actin-binding states and the swing of a distal elongated domain of myosin called the lever arm. The powerstroke is the displacement produced by the lever arm swing that in muscle allows the motor to pull on the actin track and causes shortening of the sarcomere (Fig. 1a). According to a recent unifying model, the powerstroke occurs via at least four strongly actin-bound states that occur in sequence: $P_i$ Release, Intermediate, Strong-ADP, and Rigor[10]. The rest of the motor cycle corresponds to structural states of the motor that have weak affinity for the actin filament. First, the post-rigor (PR) state is populated upon detachment from the actin filament following ATP binding. Next, the recovery stroke begins, corresponding to the transition that reprimes the lever arm, ending up in the pre-powerstroke state (PPS) during which ATP hydrolysis occurs. In the PPS, the hydrolysis of ATP is reversible, that is the nucleotide freely interconverts between ATP and ADP.$P_i$, in an approximate equilibrium. Finally, rebinding of the PPS-ADP.$P_i$ state to the actin filament triggers the transition from the PPS to the $P_i$ Release state that promotes stronger binding to the actin filament. During this transition, the active site opens to allow for phosphate ($P_i$) release[10, 11], making the step irreversible. The motor then proceeds through the strongly bound states during which the powerstroke occurs, generating tension on the actin filament to which it is bound.

The kinetic steps in the motor ATPase cycle that OM influences have been identified. OM shifts the equilibrium of the recovery stroke and the myosin ATP hydrolysis step toward the ADP.$P_i$-bound state, thus increasing the population of heads in the PPS state ready to bind to the actin filament[8]. Upon attachment to the actin filament, the rate and amplitude of phosphate release from myosin increases if OM is bound since more myosin heads can bind to the actin filament. In contrast, the rate of ADP release by myosin attached to the actin filament is not changed whether OM is present or not[4, 9].

A structure of OM bound to cardiac myosin has recently been determined in the PR state without nucleotide bound[12]—a state myosin populates only after detaching from the filament (Fig. 1a). However, the PR state is not the step in the myosin cycle that appears important for the action of OM as an activator based on the enzymology. Moreover, this previously reported binding site does not provide a rationale for the selectivity of OM for cardiac myosin, as compared to other muscle myosins.

Here, we describe a previously unseen conformation of the bovine cardiac myosin motor in the PPS state that OM stabilizes and binds to with high affinity. The drug location differs from the previously reported binding site[12]. This specific allosteric site for OM binding not only accounts for the selectivity of the drug for cardiac myosin, but it also provides a strong rationale for the mechanism of cardiac myosin activation.

## Results

**OM binds to a specific pocket to stabilize the PPS state.** To reveal the basis of the mechanism of activation by OM and its specificity, it is critical to visualize the drug in the myosin transition states that are directly affected by OM during the acto-myosin cycle. The previously reported pocket for the OM-binding site in the PR structure[12] is closed and thus unavailable in the PPS state of myosins (Supplementary Fig. 1a, b). Binding at this PR site therefore does not account for the stabilization of the motor at the end of the recovery stroke (in the PPS ADP.$P_i$ state, Fig. 1a). Kinetic studies have shown that stabilization of this state is an essential property of OM to function as an activator of force production in the sarcomere[8].

To visualize how OM stabilizes states of myosin at the end of the recovery stroke, we co-crystallized the bovine β-cardiac myosin S1 fragment in the PPS state with OM bound (OM-S1-PPS structure) at 2.45 Å resolution (using vanadate as a $P_i$ analog; Fig. 1b, c and Table 1). We were also successful in crystallizing the bovine β-cardiac muscle myosin in the PPS state without drug bound (APO-MD-PPS structure) at 3.10 Å resolution, from crystals obtained after proteolysis in situ in the presence of MgADP and inorganic phosphate ($P_i$; Fig. 1d and Table 1).

Comparison of these two cardiac PPS structures reveals that drug binding does not significantly alter the structure of the motor domain (Fig. 2a). The two motor domains of OM-S1-PPS and APO-MD-PPS structures can be superimposed overall with a root-mean-square deviation (r.m.s.d.) of 0.88 Å for 557 Cα atoms between the OM bound and APO structures and with a r.m.s.d. of 1.33 Å for 593 Cα atoms if the relay and converter are also taken into consideration. Comparison with other myosin II isoforms in the same structural state reveals that the cardiac myosin PPS state is also very similar to that of *Argopecten* (scallop)-striated muscle myosin II (1QVI[13], r.m.s.d. of 0.75 Å for 582 Cα atoms; Fig. 2b) and chicken smooth muscle myosin (SMM) II (1BR1[14], r.m.s.d. of 0.95 Å for 547 Cα atoms). When the β-cardiac OM-S1-PPS and the APO-MD-PPS structures are compared, the major differences are observed surrounding the OM-binding pocket (Fig. 2c, d), which directly account for the stabilization of a primed position for the lever arm (Fig. 1b, c).

**OM favors states with a lever arm primed.** In the OM-S1-PPS structure, OM binds in a previously unseen pocket of the motor, which we call the "PPS" allosteric site. This "PPS" pocket is not only separated by more than ~18 Å from the previously described site in the PR state[12] ("PR" pocket), but also its environment is completely different (Fig. 1e). The high quality of the electron

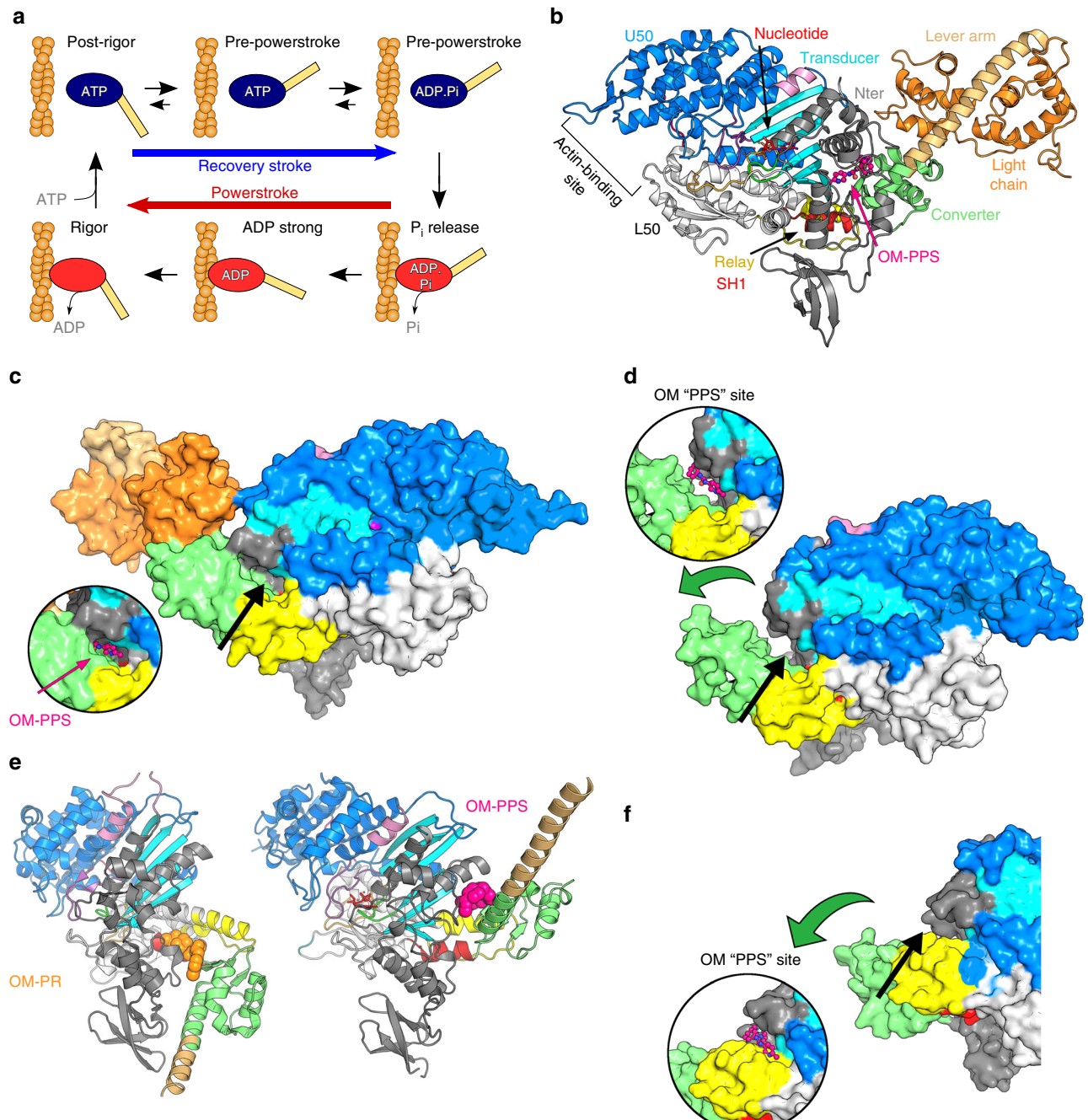

**Fig. 1** OM binds to a specific pocket of the PPS that stabilizes the lever arm primed position. **a** Myosin chemomechanical cycle. Myosin motors generate force upon releasing ATP hydrolysis products when attached to F-actin (*orange filament*). The swing of the lever arm (*yellow*) is associated with force generation (Powerstroke) when myosin is strongly bound to F-actin (myosin states in *red*). At the end of the stroke, nucleotide-free myosin is strongly attached to F-actin (Rigor). Myosin detaches from F-actin upon ATP binding and adopts the PR state. It then undergoes transitions that reprime the lever arm during the recovery stroke (*blue arrow*). ATP hydrolysis stabilizes the PPS. F-actin triggers a series of conformational changes within the motor associated with force production. The motor first populates the $P_i$ Release state without major changes in the lever arm position (PPS to $P_i$ Release transition). Then a series of conformational changes occur associated with force production (Powerstroke) and ADP release. **b** Overall view of the cardiac myosin motor domain (OM-S1-PPS structure). In the PPS state, OM (*pink*) introduces itself in a buried pocket between the N-terminal domain (*grey*), the transducer (*cyan*), the relay helix (*yellow*), and the converter (*green*). This color code is used in all figures except when indicated. **c** Surface representation of the cardiac OM-S1-PPS structure. Most of the OM molecule (*pink*) is not accessible to solvent. **d** Surface representation of the cardiac APO-MD-PPS structure. The *inset* represents the position of OM in the PPS state. Note that the OM-PPS site is not completely closed as in **c**: the converter (*green*) orientation is less primed. **e** The "PR" allosteric binding pocket for OM (*orange*) and the "PPS" binding site for OM (*pink*) are quite distant from one another. See also Supplementary Movie 1. **f** Surface representation of myosin IIb in the rigor state (4PD3[34]). The OM-PPS site is completely open and unable to bind OM strongly since the converter is unprimed. (The *green arrow* indicates the swing compared to the primed position as found in **c**). The *inset* shows the location of the OM "PPS" site. No bond can occur between OM and the converter (*green*) when it has swung in a post-stroke position

**Table 1 Data collection and refinement statistics**

| Data collection | OM-S1-PPS MgADP.VO$_4$ | APO-MD-PPS MgADP.P$_i$ |
|---|---|---|
| Space group | P2$_1$2$_1$2$_1$ | C222$_1$ |
| *Cell dimensions* | | |
| a, b, c (Å) | 98.3, 122.5, 187.4 | 87.7, 149.8, 154.3 |
| α, β, γ (°) | 90.0, 90.0, 90.0 | 90.0, 90.0, 90.0 |
| Resolution (Å) | 50–2.45 (2.54–2.45)* | 50–3.02 (3.21–3.02) |
| R$_{sym}$ | 0.30 (1.73) | 0.28 (0.94) |
| I/σ | 5.41 (0.96) | 8.35 (1.62) |
| CC$_{1/2}$ (%) | 98.6 (38.3) | 99.6 (60.2) |
| Completeness (%) | 99.3 (95.9) | 98.0 (80.0) |
| Redundancy | 8.6 (8.0) | 13.4 (4.0) |
| *Refinement* | | |
| Resolution (Å) | 24.22–2.45 | 43.39–3.10 |
| No. of reflections | 710,243 (total), 82,904 (unique) | 19,862 (total), 1585 (unique) |
| R$_{work}$/R$_{free}$ (%) | 18.12/22.34 | 24.67/31.98 |
| No. of atoms | | |
| Protein | 14,851 | 5,404 |
| Ligand/ion | 295 | 45 |
| Water | 857 | 72 |
| B factors | | |
| Protein | 56.83 | 85.87 |
| Ligand/ion | 65.22 | 56.97 |
| Water | 55.71 | 61.48 |
| r.m.s.d. | | |
| Bond lengths (Å) | 0.010 | 0.010 |
| Bond angles (°) | 1.12 | 1.23 |
| PDB code | 5N69 | 5N6A |

*Values in parentheses are for highest-resolution shell

density map for the OM-S1-PPS structure allows unambiguous placement and orientation of the drug in the PPS site (Supplementary Fig. 2). OM occupies a pocket that can form only in states of the motor with a primed lever arm. In this buried pocket, OM is at the center of a network of interactions between the N terminus, the relay helix, and the converter domain (Fig. 3). By interacting with all these elements that contribute to control the lever arm swing, OM stabilizes the PPS state, in which the lever arm is primed.

In order to confirm that the PPS state is favored by OM, small-angle X-ray scattering (SAXS) studies were performed to demonstrate that the state populated by the drug in solution is indeed the OM-S1-PPS structure we crystallized. The quality of the fit ($\chi^2 = 1.69$) agrees with the conclusion that the OM-S1-PPS structure is the conformation adopted in solution (Fig. 4a). The scattering curves indicate that the cardiac S1 fragment with OM and MgADP.VO$_4$ bound exists in a similar conformation to that adopted by the myosin motor with MgADP.VO$_4$ bound and no drug (PPS state; Fig. 4b). In contrast, the scattering curve is quite different when MgADP alone is bound to the motor. In the absence of actin, MgADP favors states with the lever arm down at the beginning of the recovery stroke such as the PR state. However, it is likely that these myosin MgADP heads can transiently explore conformations with the lever arm primed, which would allow OM binding and shift the population of conformations toward states with the primed lever arm. Thus, OM binding to the myosin motor when MgADP is present in the active site stabilizes a new conformation close to the PPS state even in the absence of the phosphate analog, VO$_4$ (Fig. 4b).

OM thus clearly stabilizes states in which the lever arm is primed (Fig. 5). These results are in full agreement with the finding that OM slows the rate of the reverse recovery stroke that unprimes the lever arm[8]. The "PPS" pocket in which OM binds requires the contribution of a primed converter (Fig. 1c). The drug pocket can only form in states at the end of the recovery stroke (PPS) or at the beginning of force generation (P$_i$ Release state (P$_i$R), Supplementary Fig. 1c), in which the converter adopts a primed position (Figs. 1c; 5). OM binding to this "PPS" site cannot occur in states of the motor with a lever arm down (Rigor and PR, Fig. 1f and Supplementary Fig. 1c) that are populated after the powerstroke or upon detachment of the motor from its track (Fig. 1a). Taken together, these findings explain how prior to the start of cardiac systole, OM increases the number of heads with the lever arm primed, which can produce force upon actin binding once calcium binds to the troponin–tropomyosin complex (Fig. 5).

**The PPS OM-binding site**. In the PPS state, OM binding involves extensive interactions with the N-terminal subdomain (K146, R147, N160, Q163, Y164, T167, and D168), the relay helix (H492), the extremity of the third beta-strand of the transducer (H666), and the converter (P710, N711, R712, I713, R721, Y722, F765, L770, and E774), as depicted in Fig. 3. The nature of the interactions is predominantly hydrophobic, although the methyl-pyridinyl ring forms a polar contact and the carbamoyl-amino moiety makes five polar contacts with both side chains and main chain atoms of the surrounding residues. Most of the OM molecule is buried in the structure, and only the methyl-pyridin-3-yl ring is partially exposed to the solvent (Fig. 1c), which is consistent with the tolerance to substitutions made at this site during the optimization effort that produced OM[15]. The characteristics of this "PPS"-binding pocket are compared to that of the previously reported "PR"-binding site for OM in Supplementary Table 1. In particular, the OM drug is more tightly bound in the "PPS" site, as indicated by the electron density map (Supplementary Fig. 2), and the fact that apo crystals can't be grown spontaneously in conditions where OM-S1-PPS crystals grow, arguing in favor of a full occupancy of the PPS site by OM. In contrast, the occupancy of the "PR" site by OM is lower as shown by local ligand density fit (LLDF) values (Supplementary Table 1) and this correlates with isothermal titration calorimetry (ITC) measurements (Table 2 and Supplementary Fig. 3). The K$_d$ for OM binding to cardiac myosin is 0.29 μM when the motor is bound to nucleotide analogs that favor the PPS state. In contrast, no binding is detected when the motor is depleted of nucleotide, in which case the motor mostly explores PR and rigor-like states.

Upon binding to the cardiac motor domain, OM adopts a crescent shape with a sharp bend (Fig. 2). Interestingly, comparison of the two OM-S1-PPS molecules present in the asymmetric unit reveals that both OM molecules are in the exact same position and interact similarly, consistent with the fact that the drug is tightly and specifically bound. Comparison of the structures with and without drug bound reveals that the OM-binding pocket is actually not completely formed in the APO-MD-PPS structure (Fig. 1c, d). Tight binding of OM in the "PPS" pocket stabilizes a specific position of the primed lever arm among those explored by cardiac myosin in the PPS state (Figs. 1b, c and 2). Interestingly, in the presence of OM, the lever arm adopts a primed lever arm position that is close to that previously observed for scallop myosin II (Fig. 2b), while the cardiac apo PPS structure has a lever arm position slightly less primed, with concerted changes in the relay and the converter position (Fig. 2c, d). In summary, the drug binds into a pocket via induced fit and consequently stabilizes the lever arm in a fully primed position.

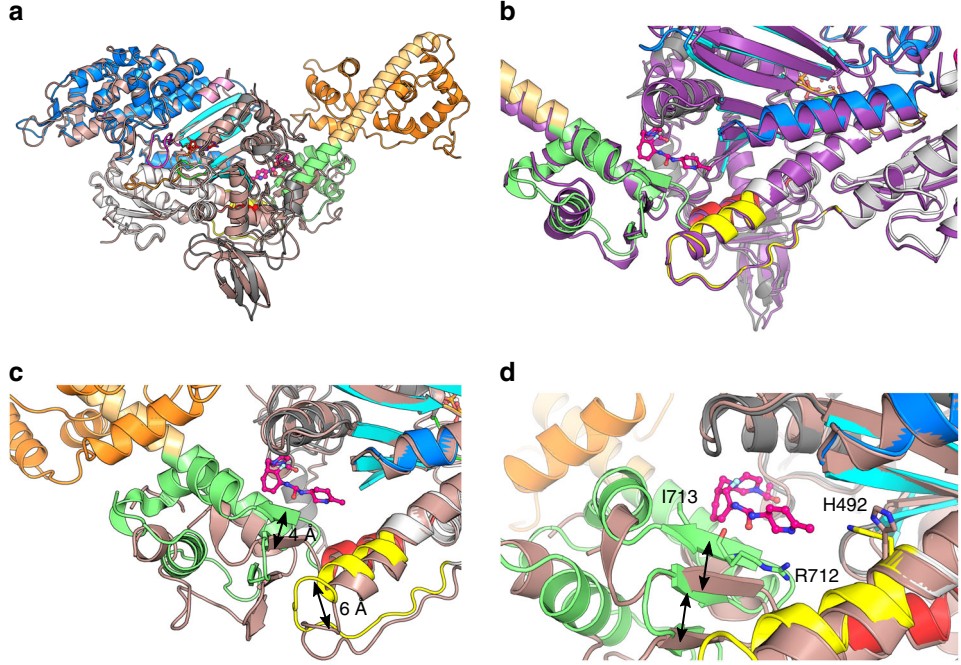

**Fig. 2** Binding of OM stabilizes a specific primed position of the myosin lever arm and does not affect the motor domain conformation. **a** The features of the motor domain including the position of the subdomains (Nter (*grey*), U50 (*blue*), L50 (*white*)) and key connectors are similar for the OM-S1-PPS (*multi-colored*) and the APO-MD-PPS (*pink*) structures. **b** Superimposition of the OM-S1-PPS (*multi-colored*) structure with the *Argopecten* skeletal myosin II in the PPS state (1QVI[13], *purple*). **c** Comparison of the relay and the converter in the APO-MD-PPS structure (*pink*) and the OM-S1-PPS (*multi-colored*). The relay (*yellow*) is displaced by ~ 6 Å, and the converter subdomain (*green*) is rotated (15°) and translated (~ 4 Å). Overall, OM binding stabilizes a conformation of the converter that forms a specific "PPS" allosteric binding site for OM and maintains the converter in a primed position. **d** Detail of the OM "PPS" binding pocket. Closure of the allosteric site (*black arrows*) stabilizes interactions between OM and converter residues, such as Arg712 and Ile713 as well as His492 which adopts a new conformation to interact with OM

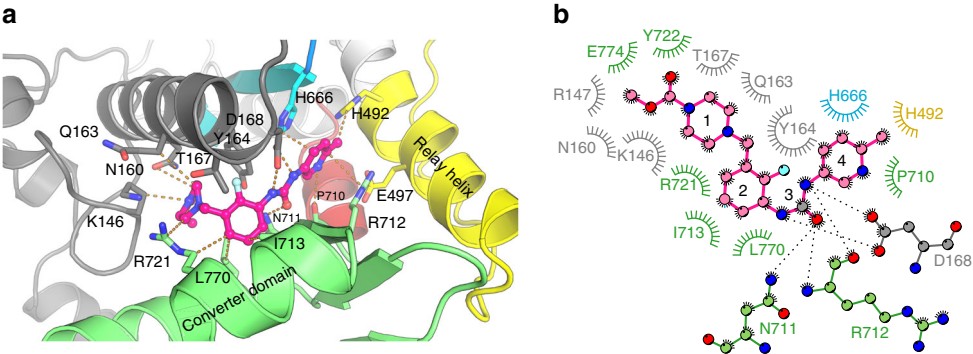

**Fig. 3** The OM PPS binding site in the cardiac myosin motor domain. Ribbon **a** and diagram **b** representations of the most extensive interactions found in the OM "PPS" binding pocket. The different regions of the OM molecule are indicated with numbers: 1 carboxymethyl-piperazine; 2 fluoro-benzyl ring; 3 carbamoyl-amino linker; 4 methyl-pyridinyl ring

**The PPS binding site of OM accounts for its specificity.** The "PPS" allosteric OM-binding site found in OM-S1-PPS agrees with the previously reported OM-binding site identified using a photo-reactive benzophenone derivative of OM that crosslinked to Ser148[4] (Supplementary Fig. 1d). The orientation and position of the drug described in the "PR"-binding site[12] (Supplementary Fig. 1d) are not compatible with crosslinking of this OM derivative to Ser148. Together, this analysis strongly supports the conclusion that the "PPS"-binding site revealed with the OM-S1-PPS structure is the critical location OM occupies to increase the force produced by β-cardiac myosin.

Additionally, the network of interactions between the drug and the motor in the "PPS"-binding site readily explains the specificity of OM action toward cardiac myosin and not other closely related myosins, such as SMM and skeletal muscle myosin[4] (Table 3). The most important packing interactions with OM are accomplished with residues Y164, D168, H492, H666, N711, R712, and I713, which are strictly conserved between bovine and human cardiac myosin (Table 3). Interestingly, the majority of these residues are not conserved in the sequence of other myosin II isoforms (Table 2). In particular, four main residues vary in these sequences: Y164 (Ser in SMM, Phe in Skeletal), H666 (Thr in SMM), N711 (Ser in Skeletal), and I713 (Val in Skeletal). These important differences plainly account for the selectivity of OM for cardiac muscle myosin II. In contrast, most of the residues found in the "PR" pocket are conserved among myosin II isoforms (Supplementary Table 2).

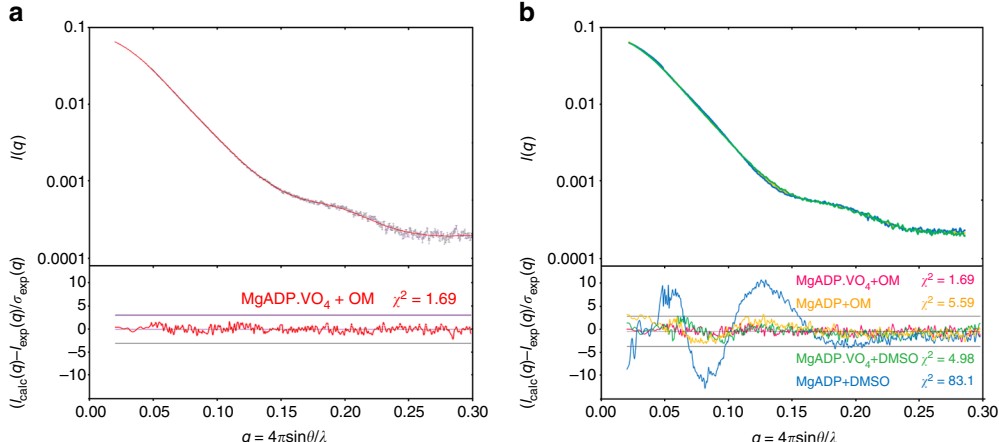

**Fig. 4** SAXS data show that, in solution, OM populates a myosin conformational state similar to PPS. **a** (*Upper panel*) Experimental scattering intensities of the MgADP.VO$_4$+OM condition in *black* (with associated *error bars*) superimposed on the calculated scattering patterns of the OM-S1-PPS structure is shown as a continuous *red line*. (*Lower panel*) Reduced residuals of the least-squares fits shown on a linear scale ($\pm 3\sigma$ indicated with *horizontal lines*). The quality of the fit ($\chi^2 = 1.69$) reveals that the crystal structure corresponds to the conformation adopted in solution. **b** (*Upper panel*) Experimental scattering intensities of the MgADP+OM (*yellow*), MgADP.VO$_4$+DMSO (*green*), and MgADP+DMSO (*blue*). (*Lower panel*) The $\chi^2$-value reflects the discrepancy between the different experimental curves and the theoretical scattering of the OM-S1-PPS structure. Note how similar are the curves with MgADP.VO$_4$ or bound to OM, and how they differ from that of MgADP in the absence of OM (*blue*)

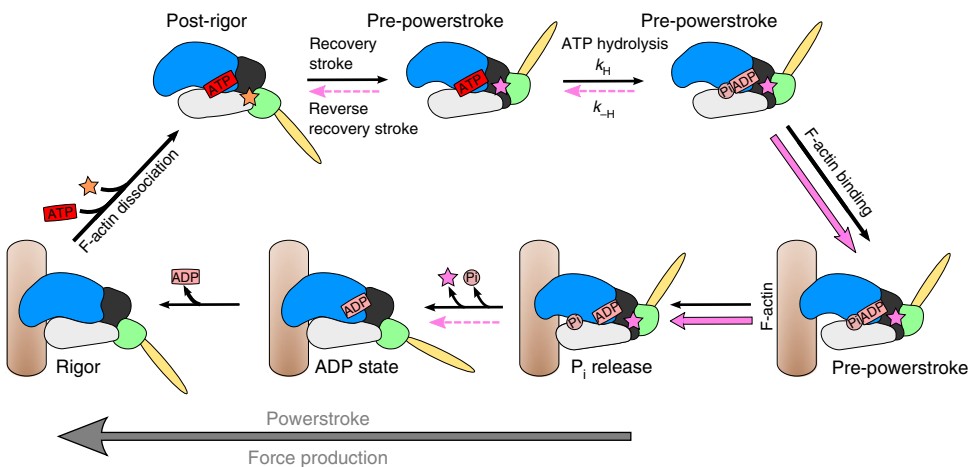

**Fig. 5** Effect of OM on the chemomechanical actomyosin ATPase cycle. The motor cycle is presented, highlighting the conformational changes that the motor domain undergoes along the motor cycle as well as the states to which OM can bind in either the "PR" site (*orange star*[12]) or the "PPS" site (*pink star*). ATP hydrolysis stabilizes the Pre-powerstroke state, as does the binding of OM to the "PPS" site, which greatly slows the reversal of the recovery stroke (*dotted pink lines*). Rebinding to F-actin (brown filament) in this PPS state triggers a series of conformational changes associated with force production (Powerstroke). OM increases the number of heads ready to produce force (*red square*). OM binding in the "PPS" site increases the rate of the transition that allows the release of P$_i$ (*pink arrows*) since this step occurs while the lever arm stays primed. The following transition destabilizes the "PPS"-binding pocket since it requires the swing of the lever arm. Time-resolved FRET studies[18] have shown that this step is slowed (*dotted pink arrow*). OM thus increases force production by increasing the number of myosin heads bound to F-actin in the early stages of force production

**OM dissociates from myosin during the recovery stroke**. The two "PR"-binding sites and "PPS"-binding sites of OM are far away from one another in the pre-recovery and post-recovery stroke structural states (Fig. 1d). This raises the question of whether the drug could remain bound while the motor undergoes the recovery stroke as some binding interactions may not be maintained. In fact, a number of observations indicate that the relocation of the drug from the "PR" to the "PPS" site may involve drug dissociation. First, the residues that are common in the two binding sites (P710, N711, R712, and L770) make drastically different interactions with the drug in the two binding sites (Supplementary Fig. 1d). Moreover, the orientation of the drug is opposite in the two sites (Supplementary Figs. 1d and 4).

If the OM concentration is sufficient to bind in the "PR" pocket, the 180° rotation required for the drug in such a narrow pocket indicates that detachment from the site is likely upon the recovery stroke.

**PPS structure sheds light on the mechanism of action of OM.** As the "PPS" OM-binding pocket closes, it buries the drug allowing OM to act as a bridge between the converter domain, the relay helix, and the N-terminus subdomain near the third beta-strand of the transducer. By doing this, OM stabilizes the PPS and thus favors ATP hydrolysis, increasing the number of heads with ADP.P$_i$ at the end of the recovery stroke, ready to

**Table 2 ITC-binding experiments**

| Parameter | Nucleotide state | | | | |
|---|---|---|---|---|---|
| | AMPPNP (n = 5) | ADP-BeFx (n = 5) | ADP-VO$_4$ (n = 5) | ADP (n = 5) | Nucleotide-free (n = 5) |
| Stoichiometry (N) | 0.81 ± 0.01 | 0.79 ± 0.02 | 0.80 ± 0.01 | 0.37 ± 0.06 | ND |
| Affinity ($K_d$ μM) | 2.80 ± 0.37 | 1.80 ± 0.21 | 0.29 ± 0.04 | 5.3 ± 1.8 | ND |
| Enthalpy ($\Delta H$, kcal mol$^{-1}$) | 9.40 ± 0.63 | 8.40 ± 0.46 | −3.50 ± 0.12 | 7.4 ± 2.9 | No heat signal |
| Entropy ($\Delta S$, e.u.) | 59.0 ± 2.0 | 56.0 ± 1.4 | 17.0 ± 0.5 | 50.0 ± 9.8 | ND |

OM binding to cardiac myosin is nucleotide-dependent and is stronger in the presence of MgADP.VO$_4$ in the binding pocket. The ability for myosin to adopt states with a primed lever arm depends on the nucleotide in the following order (Nucleotide free < ADP < AMPPNP < ADP.BeF$_x$ < ADP.VO$_4$), which is the same as that found for the affinity of OM to cardiac myosin. Averaged means for five technical replicates ± SD are shown

commit to binding the actin filament once cardiac systole begins. Interestingly, the stabilization of this PPS state with ADP.P$_i$ bound by OM also enhances the trapping of P$_i$ in this state as shown by the reduced ATPase activity in the absence of F-actin when OM is bound[4, 8, 9]. This trapping of P$_i$ until F-actin binds is absolutely essential for motor activity[16]. In contrast, once these primed PPS myosin heads bind to the actin filament, P$_i$ is rapidly released[4, 8], indicating that OM binding in the "PPS" pocket is compatible with the PPS to P$_i$ release transition (Supplementary Fig. 1c).

## Discussion

Myosin motors are complex machines that must undergo a series of precise and timely controlled transitions for their activity. In cardiac muscle, cardiac myosin comprises, in part, the thick filament of the cardiac sarcomere and powers cardiac contractility. To test the hypothesis that directly increasing cardiac performance might benefit patients with HF-rEF, selective, direct activators of cardiac myosin were identified by high-throughput screening of a synthetic small-molecule library using a reconstituted cardiac sarcomere assay[4]. Several chemical classes that directly activate cardiac myosin force production in the sarcomere were discovered, and one compound class was optimized extensively using an iterative process. OM, a selective, small-molecule cardiac myosin activator is the most advanced exemplar of this novel mechanistic class and, after extensive preclinical and clinical study, is now being tested in a large Phase 3 cardiovascular outcomes trial in patients with HF-rEF (GALACTIC-HF, www.clinicaltrials.gov identifier, NCT02929329).

Unlike small-molecule inhibitors of motor function, an activator must interact with the motor, cardiac myosin in this case, in a manner that does not stop the motor from cycling. While the manner in which OM affects the ATPase cycle of cardiac myosin has been described, to date, no study has revealed the precise structural mechanism that allows this activator to increase the contractile force generated by the cardiac sarcomere. Considering the complexity of the rearrangements that the myosin motor undergoes through the ATPase cycle, the mode of action of an activator to increase the force generated by the sarcomere is not straightforward.

In cardiac muscle, the myosin motor cycles at a rate that allows it to undergo only one or two crossbridge cycles during the contractile period of a single heartbeat[17]. Further, during systole only about 10–30% of the cardiac myosin heads engage the actin filament to produce force[17]. It is clear that a cardiac myosin activator must not significantly slow down several steps of the motor cycle (such as hydrolysis, motor attachment to F-actin, and detachment from F-actin), otherwise the ability of the motor to progress through its cycle would likely be impaired. On the other

hand, speeding up the motor sufficiently to allow it to undergo more crossbridge cycles during a single heartbeat is probably not feasible, given that one would need to accelerate the cycle substantially to do so. OM appears to take advantage of another means of increasing contractile force; it increases the number of myosin heads that engage the actin filament during the cardiac contraction.

The cardiac structure with OM bound presented here has elucidated the main principles that allow OM to increase the force generated by the cardiac sarcomere (Fig. 5). Prior to the start of contraction, the myosin head can exist in a number of states, some of which cannot bind to the actin filament. OM stabilizes the PPS state of the motor in which the lever arm is primed and ready to bind to actin filament. Thus, there is a larger population of heads with a primed lever arm ready to bind to the actin filament and contribute to force once calcium binds to the troponin–tropomyosin complex to initiate the contractile cycle. Further, by stabilizing this PPS state, OM slows the non-productive turnover of ATP, meaning the release of ADP and P$_i$, in the absence of a force producing interaction with the actin filament.

The mechanism revealed here for OM based on the visualization of its binding site is in agreement with the current model we have proposed in which P$_i$ is released from a state with the lever arm still primed[10, 11]. This mechanism explains why OM increases the rate of Pi release for myosin when OM is bound. The structural data also suggest that OM is released from the motor when the lever arm swings, since the "PPS" pocket opens as the powerstroke proceeds; consistent with this model is the finding that the binding affinity of OM is highest for the PPS state (0.29 ± 0.04 μM) and more than 20-fold lower for states in which the lever arm is down. The fact that OM would not be associated with the motor at the end of the powerstroke is consistent with the fact that the last step of the powerstroke (namely ADP release) is not affected by OM[4, 8, 9].

Recent time-resolved Förster resonance energy transfer (FRET) studies[18] under high OM concentrations indicate that after fast Pi release, OM slows the next transition, which is associated with the lever arm swing, resulting in an increase of the duty ratio[9]. Our model predicts that OM binding to the "PPS" pocket could indeed slow the transitions that result in opening this site, in particular those that require a lever arm swing. Higher duty ratio also leads to the increase in calcium sensitivity and slow force development in cardiac muscle that have been reported upon experiments with high OM concentrations (10 μM)[9]. However, given that at clinical doses the effective unbound concentration of OM in human plasma[3, 15] (<200 nM) is similar to the $K_d$ for heads with MgADP.VO$_4$ bound and lower than the binding affinity of OM to any other nucleotide state, the findings at high concentrations of OM may not be relevant to its effect on myocardial contractility. Further, since only a fraction of the myosin

**Table 3 Sequence variability for myosin residues found in the OM PPS binding pocket**

| Residue number # | Gene | Uniprot | N-ter cavity (143 148) | N-ter cavity (160 — 170) | Relay helix (492 497) | Transd. (666) | Converter domain (710 — 722) | Converter (770) |
|---|---|---|---|---|---|---|---|---|
| Hs-βCar-Myo2 | *MYH7* | P12883 | GKKR | NAYQYMLTDRE | HMFVLE | H | PNRILYGDFRQRY | L |
| Bt-βCar-Myo2 | *MYH7* | Q9BE39 | . . . . | . . . . . . . . . . . | . . . . . . | . | . . . . . . . . . . . . . | . |
| Hs-αCar-Myo2 | *MYH6* | P13533 | . . . . | . . . . . . . . . . . | . . . . . . | . | . . . . . . . . . . . . . | . |
| Hs-Sm-Myo2 | *MYH11* | P35749 | . . . . | T..RS..Q... | T..... | T | ....VFQE..... | . |
| Gg-Sm-Myo2 | *MYH11* | P10587 | . . . . | T..RS..Q... | T..... | N | ....VFQE..... | . |
| Rb-Sk-Myo2 | *MYH4* | Q28641 | . . . . | ....F...... | . . . . . . | S | .S....A..K... | . |
| Rb-Sk-Myo2 | *MYH13* | Q9GJP9 | . . . . | ....F.....D | . . . . . . | . | .S....A..K... | . |
| Gg-Sk-Myo2 | *N116* | P13538 | . . . . | ....F...... | . . . . . . | . | .S.V..A..K... | . |
| Hs-NMM2a | *MYH9* | P35579 | . . . . | T..RS.MQ... | T..I.. | N | ...VVFQE..... | . |
| Hs-NMM2b | *MYH10* | P35580 | . . . . | S..RC..Q... | T..I.. | N | ....VFQE..... | . |
| Hs-NMM2c | *MYH14* | Q7Z406 | . . . . | G..RS..Q... | T..... | N | .....FQE..... | . |
| Hs-Myo5a | *MYO5A* | Q9Y4I1 | .QNM | E..KQ.AR.ER | .V.K.. | T | .S.WT.QE.FS.. | V |
| Hs-Myo6 | *MYO6* | Q9UM54 | ..SL | K.FRD.KVLKM | RILKE. | G | .S.ASFHELYNM. | F |
| Hs-Myo10 | *MYO10* | Q9HD67 | RRHL | EC.RCLWKRHD | .I.S.. | N | AV.RPFQ..YK.. | E |

Sequence comparison of the residues found in the OM "PPS"-binding site in different myosin family members. The residues directed toward the OM-binding site are highlighted in *red* and differ in sequence among myosin IIs. *Dots* (.) indicate identical residues. The Hs, Bt, Gg, and Rb abbreviations stand for human, bovine, chicken and rabbit myosins, respectively. Note that OM does not influence the activity of chicken gizzard smooth muscle myosin, *Myh11*[4], rabbit psoas muscle fast skeletal myosins, *Myh4* and *Myh13*[4], and chicken skeletal muscle myosin, *N116*[12].

heads in the sarcomere are bound to OM, the structure of the thick filaments could allow OM-free heads to accelerate the lever arm swing of OM-bound heads, thus minimizing any change to the kinetics of the lever arm swing. Importantly, our structures indicate that OM does not change the conformations that the motor domain explores during the motor cycle but acts by increasing the stability of the states with lever arm primed. The motor mechanism per se is unchanged (see also Supplementary Note 1). Thus, it seems likely that OM modulates force production mainly by increasing the number of heads ready to undergo a powerstroke during systole.

Structural studies of myosin motors with allosteric drugs provide an elegant approach to gain new insights on the powerstroke mechanism itself as well as its specific modulation. While molecular motors are particularly complex machines, the principles that distinguish activators from inhibitors certainly apply to other molecular machines. In the coming years, new modes of action to alter the force produced by these motors may emerge[19]. Specific modulators of myosin have great potential to result in new treatments against diseases of human muscle[3, 19, 20] as is now being tested in patients with heart failure.

## Methods

**Purification of cardiac myosin.** Full-length cardiac myosin was prepared from bovine hearts (Pel-freez Biologicals) via the method of Margossian et al.[21], drop-frozen in liquid nitrogen, and stored at −80 °C. Subfragment-1 was prepared by limited chymotryptic digestion based upon the method of Weeds and Taylor[22]. Full-length myosin was precipitated by >10-fold dilution in low-salt buffer (12 mM K-Pipes, 2 mM MgCl$_2$, 1 mM dithiothreitol (DTT), pH 6.8), pelleted by centrifugation (5000 × *g*, 30 min, 4 °C), and the resulting myosin filaments were resuspended in digestion buffer (20 mM K-Pipes, 10 mM K-EDTA, 1 mM DTT, pH 6.8). Myosin was digested in a filamentous form by addition of tosyl-L-lysyl-chloromethane hydrochloride (TLCK)-treated α-chymotrypsin (Sigma), followed by incubation at 22 °C for 30 min with occasional mixing. Digestion was terminated by addition of phenylmethylsulfonyl fluorid (PMSF) (1 mM final), and the soluble S1 fraction was separated from insoluble myosin rods by centrifugation (29,000 × *g*, 30 min, 4 °C). Cardiac S1 was precipitated using ammonium sulfate (60% w/v final) and isolated by centrifugation (29,000 × *g*, 30 min, 4 °C). The resulting S1 pellet was resuspended and dialyzed against two to three changes of low-salt buffer (12 mM K-Pipes, 2 mM MgCl2, 1 mM DTT, 0.1 mM PMSF, pH 6.8) before being clarified by centrifugation (142,000 × *g*, 2.5 h, 4 °C). This intermediate S1 fraction was stabilized by addition of sucrose (10% w/v) before being drop-frozen in liquid nitrogen and stored at −80 °C. For crystallography experiments, this intermediate S1 fraction was further purified by anion-exchange chromatography on Mono-Q (GE Healthcare) in 20 mM Tris, 0.8 mM NaN$_3$, pH 8.0 (at 4 °C) using a 0–350 mM gradient of NaCl. Target fractions were pooled and buffer-exchanged into crystallization buffer (10 mM HEPES, 50 mM NaCl, 1 mM NaN$_3$, 2.5 mM MgCl$_2$, 0.2 mM ATP, 1 mM TCEP, pH 7.5) by repeated concentration using 15 kDa MWCO Amicon Ultra centrifugal concentrators (EMD Millipore). The final S1 (20–30 mg ml$^{-1}$) was supplemented with MgADP to a final concentration of 2 mM, aliquoted into cryotubes, and flash-frozen in liquid nitrogen prior to storage at −80 °C for later use.

**Isothermal titration calorimetry.** Isothermal titration calorimetry experiments were carried out using a Micro-Cal Auto ITC HT microcalorimeter (Microcal Inc., now Malvern, Inc.) at 10 °C. A solution of 300 μM OM in 12 mM PIPES (pH 6.8), 2 mM MgCl$_2$, 5 mM β-mercaptoethanol, and 3% v/v dimethyl sulfoxide (DMSO) (pH 6.8) was titrated into the sample cell, which contained 20 μM bovine cardiac myosin S1 in the same buffer. Nucleotides and nucleotide analogs were included in both titrant and myosin sample at 2 mM. For the nucleotide-free condition, myosin sample contained 17 μg ml$^{-1}$ apyrase (Sigma). The S1 concentration was determined by ultraviolet absorbance (280 nm) in 6 M guanidine-HCl using a calculated extinction coefficient of 95,000 M$^{-1}$ cm$^{-1}$ based on the sequence of bovine cardiac myosin (Uniprot Q9BE39, AA 1–840) and bovine MYL3 (Uniprot P85100). Injections (10 μl) were made every 300 s. To correct for the heats of dilution and slight buffer mismatches between the titrant and sample, the average heat signal from the last three injections at the end of the experiment (when binding was saturated) was subtracted from all values. Data collection and analysis was performed using the modified Origin software included with the instrument, using a single binding site model.

**Crystallization and structure determination**. The APO-MD-PPS crystals were obtained using the hanging drop vapor diffusion method at 290 K by mixing purified bovine Cardiac Myosin S1 fragment at 20 mg ml⁻¹ (170 μM) pre-incubated with 2 mM MgADP for 30 min on ice. Drops were set up by mixing 1 μl of the protein and 1 μl of the reservoir solution containing 13% PEG 3350 (w/v), 5% Tacsimate pH 6.0, 5 mM TCEP, 10% Glycerol, and 3.3% DMSO. The crystals appeared after 21 days using the micro-seeding technique. Note that the crystals are hard to reproduce as they require spontaneous/in situ proteolysis to grow (likely by the α-chymotrypsin used in the purification step) to produce a MD fragment (cleavage of the heavy chain after the converter). Crystals were cryo-cooled in liquid nitrogen in a final solution containing 15% (w/v) PEG 3350, 5% Tacsimate pH 6.0, 5 mM TCEP, 25% Glycerol, and 3.3% DMSO. X-ray data sets were collected at 100 K at the Proxima 2 A ($\lambda$ = 0.9762 Å, SOLEIL synchrotron, France).

The OM-S1-PPS crystals were obtained using the same technique. The same protein was incubated at 5 mg ml⁻¹ (43 μM) with 2 mM MgADP for 30 min, 5 mM OM for 1 h, and 2 mM vanadate for 30 min at 298 K. Crystals were obtained in 24% (w/v) PEG 3350, 5% Tacsimate pH 6.0, 5 mM TCEP, 20% Glycerol, and 10% DMSO at 298 K. The optimized crystals appeared overnight using the micro-seeding approaches (note that in this case, the whole S1 (without proteolysis) crystallizes). Crystals were flash-frozen in the crystallization condition. X-ray data sets were collected at 100 K at the ID23-2 beamline ($\lambda$ = 0.8729 Å, ESRF synchrotron, France).

The diffraction data sets were indexed and scaled with XDS[23]. Molecular replacement solution was obtained with Molrep[24, 25], using the *Argopecten*-striated muscle myosin II PPS structure (1QVI[13]) as a search model. The region of the converter and the lever arm were excluded from the search model. These regions were subsequently built in electron density using buccaneer[26]. Model building and refinement were carried out with Coot[27] and BUSTER[28], respectively. The statistics for favored, allowed and outlier Ramachandran angles are 99.24, 99.67, and 0.33% for the OM-S1-PPS structure, and 82.12, 96.28, and 3.72% for the APO-MD-PPS structure, respectively. Before deposition to the PDB, the structure were submitted to the PDB_REDO server[29]. The coordinate and geometry constraints files for the ligand were created with Coot and phenix.elbow[30]. Note that the bovine and human *MYH7* motor domains share 95.9% identity and 98.2% similarity. Out of 830 residues included in the motor domain, the main differences in sequence are found for 13 solvent-exposed residues. On the basis of this, no difference is to be expected in the structure of human and bovine cardiac myosins and this is supported by the similarity of their kinetic properties as previously reported (Deacon et al.[31]—see Table 1; Liu et al.[8]—see Table 2) in which human and bovine myosins were directly compared. Figures and movies were made using PyMol[32].

**SAXS experiments**. SAXS data were collected on the SWING beamline (synchrotron SOLEIL, France). Purified bovine Cardiac Myosin S1 was extensively dialyzed against 10 mM HEPES pH 7.5, 50 mM NaCl, 1 mM NaN₃, 2.5 mM MgCl₂, 2 mM ADP, 1 mM TCEP (without any ATP) in order to remove all the Pᵢ present in the solution. The protein was then subsequently incubated with either 5 mM OM or 10% DMSO for 1 h on ice, and then with 2 mM vanadate when necessary. All samples were centrifuged at 20,000 × $g$ for 10 min at 4 °C prior to the analysis. 40 μl of the protein at 2 and 5 mg ml⁻¹ (17 and 43 μM, respectively) were injected between two air bubbles using the auto-sampler robot. Thirty-five frames of 1.5 s exposure were averaged and buffer scattering was subtracted from the sample data. As both 2 and 5 mg ml⁻¹ curves displayed no traces of aggregation, only the 5 mg ml⁻¹ curve was used for further analysis because of the higher signal/noise ratio. As the APO-PPS-MD structure lacks the converter and the light chain, a chimeric model was built using the motor domain from the APO-PPS-MD structure, and building the lever arm helix and the light chain from the OM-S1-PPS structure using the converter position as a reference to position the lever arm. The theoretical SAXS curves were calculated with CRYSOL[33] and compared based on the quality of their fits against the different experimental curves.

**Data availability**. The atomic coordinates and structure factors have been deposited in the Protein Data Bank, www.pdb.org, with accession numbers 5N69 (OM-S1-PPS) and 5N6A (APO-MD-PPS). The SAXS data with OM has been deposited in the Small Angle Scattering Biological Data Bank (SASBDB), www.sasbdb.org, with accession numbers SAS299 (MgADP.VO₄+OM) and SAS300 (MgADP+OM).

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

## Acknowledgements

We thank beamline scientists of PX2A (SOLEIL synchrotron) for excellent support during data collection. V.J.P.-H. is the recipient of a fourth year PhD fellowship from Ligue contre le cancer. J.R.-P. is the recipient of an Association Française contre les Myopathies (AFM) fellowship 18423. A.H. was supported by grants from CNRS, FRM DBI20141231319, ANR 13-BSV8–0019–01, AFM 17235, and Ligue Contre le Cancer RS16. The A.H. team is part of the Labex CelTisPhyBio:11-LBX-0038, which is part of the IDEX PSL (ANR-10-IDEX-0001–02 PSL).

## Author contributions

A.H. designed research; V.J.P.-H. crystallized, solved the crystal structures, and performed SAXS studies; J.J.H. performed in vitro functional assays; all authors discussed and analyzed the data; A.H. and F.I.M. wrote the manuscript with the help of the other authors.

## Additional information

**Competing interests:** J.J.H. and F.I.M. are current employees and shareholders of Cytokinetics Inc. The remaining authors declare no competing financial interests.

