## [Peer review file · Nature Communications]

Reviewers' comments:

Reviewer #1 (Remarks to the Author):

This is very nice, technically well executed study presenting the structure of the small molecule cardiac myosin activator, omecamtiv mecarbil, bound to cardiac myosin. The structure was found to be locked into the primed state, with OM bound in a pocket (PPS) distinct from that previously observed. This is relevant as the primed structure is the physiologically relevant one for the explaining the kinetic effect of OM on myosin. A detailed analysis of the binding pocket explains why OM is specific for cardiac muscle myosin II. Together, these results significantly advance understanding of the mechanism by which OM activates cardiac myosin, that is by increasing the number of myosin heads bound to actin and prepared to make a power stroke.

My only comment is that the authors should include a discussion of a study on OM recently published in JBC (Sweson et al. Omecamtiv Mecarbil Enhances the Duty Ratio of Human β -Cardiac Myosin Resulting in Increased Calcium Sensitivity and Slowed Force Development in Cardiac Muscle.)

Reviewer #2 (Remarks to the Author):

Omecamtiv Mecarbil (OM) is in phase 3 clinical trials as a treatment for chronic heart failure. It was discovered more than 5 years ago but despite extensive studies its mode of action remains poorly understood. It binds to the myosin motor domain; it is specific for β -cardiac myosin and it has been shown to increase cardiac output in cardiomyocytes, hence its potential for the treatment of heart failure.

There have been three reports over the last year on the effects of OM on the ATPase cycle of human beta cardiac myosin. Yet the mechanism of action remains poorly defined. A crystal structure of human myosin with OM bound was published in Nat Commun a year ago which placed the OM in a region between the converter domain and the relay helix.

The new report using bovine β -cardiac myosin finds the position of the OM close-by but in a quite distinct conformation (of the motor and the drug) to that previously reported. The motor domain here is in the Pre Power-Stroke conformation favoured by an ADP.Pi containing myosin motor in which the myosin is poised ready to bind actin and undergo a power-stroke. The OM position is inverted compared to the earlier structure. An important feature of the model is that the location is only available at certain points in the ATPase cycle requiring the drug to bind and dissociate during each ATPase cycle.

The crystallography is beautifully presented as expected from this very experienced team. The arguments are logically presented and easy to follow. The correction to the earlier structure is important in this active field of research and the structure may help to understand the mode of action of this potentially significant human drug. However, beautiful the structures the explanations for the mode of action of OM appear to be incomplete or speculative and further information is needed before we can begin to understand how the drug produces all of the effects reported for it. This does not detract from this important work which is an essential step of the way to understanding this drug

1. The fact that this work used bovine myosin needs to be more upfront. It is not mentioned in the abstract, title or introduction.

2. Since the earlier work did use a different source of then the different binding modes of OM could be due to species differences. The conservation of the binding site between human and bovine is well demonstrated in Table 2. Is there any explanation for the different conformations of bound OM in the two structures.

3. The work provides a structural explanation for the previously reported result that OM stabilizes

the M.ADP.Pi complex at the expense of the M.ATP complex a change in the equilibrium constant for the hydrolysis step on myosin.

4. The argument is presented that OM stabilizes the PPS state by making significant contacts across the converter/ relay helix and N- terminal domains. Thus making more myosin heads available to bind actin from the M.ADP.Pi state – at the expense of the M.ATP pre recovery stroke state. This seems reasonable but this implies that OM also stabilize this M.ADP.Pi state against the following steps that destroy this OM binding pocket (reverse recovery stroke). This is observed for myosin in the absence of actin where Pi release is inhibited. But no such effect is observed in the presence of actin where Pi and is accelerated while the Vmax of the ATPase and the motility is inhibited. The same forces that act to stabilizing the M.ADP.Pi state compared to M.ATP must also resist its loss and would provide the mechanism to inhibit the overall cycling speed reported for OM. However this mechanism would also not be simply reconciled with an increase in duty ratio and higher steady state force and accelerated Pi release. The new structural data does not explain this mechanism.

5. The authors argue that their most recent model of a cycle Pi release before a power-stroke/lever arm swing is compatible with the proposed mode of action of OM. This is compatible with OM inhibiting the power stroke but there is no explanation of how the model can explain the actin induced acceleration of the Pi release step. The model of Rohde et al appears equally probable.

6. At the moment the model appears to require simultaneous release of OM and ADP at completion of the lever arm swing. Is this possible? If the lever arm swing allows both events then OM released like ADP release will be load dependent. Inhibiting the lever arm swing would give higher force in the steady-state but also slow the cycle and ADP release under zero load.

7. It seems that a more holistic look at the cycle and where OM might affect the cycle with and without load will be needed before we understand the role of OM

8. If the PPS, M.ADP.Pi state is stabilized it might be predicted stabilize the "J-motif" or "off-state" of the thick filament making it harder to activate the thick filament –leading to lower force. REF Hooijman P, Stewart MA, Cooke R (2011) Biophys J 100:1969-1976.

9. This M.ADP.Pi state is also proposed to be the form of myosin interacting via the myosin mesa with MyBP-C. The OM could interfere alter the MyBP-C interaction.

10. If OM does stabilize the M.ADP.Pi state at the expense of M.ATP the drug may be useful in correcting the problem with HCM mutations in which the mutation inhibits the ATP hydrolysis step thus favoring M.ATP over M.ADP.Pi. i.e. the classic R453C mutation associated with severe HCM.

Reviewer #3 (Remarks to the Author):

Planelles-Herrero et al. describe the crystal structures of cardiac myosin in the pre-powerstroke state (PSS), both with (2.45 Å) and without (3.0 Å) the selective activator Omecamtiv mecarbil (OM) bound. This allows them to derive mechanism of action of OM. The binding of OM binding does not significantly alter the structure of the motor domain. Instead it binds to an allosteric site so that the lever arm rests in a primed position increasing the population of heads in the PPS state ready to bind to the actin filament. This is especially interesting as previously a different binding site of OM was described by Winkelmann et al 2015 (in Nature Communication). However, this structure did not provide a rationale for the selectivity of OM for cardiac myosin, as compared to other muscle myosins. In contrast, the new binding site not only accounts for the selectivity of the

drug for cardiac myosin but it also provides a strong rationale for the mechanism of cardiac myosin activation which is in strong agreement with previous kinetic studies.

The data seems sound and the conclusions are well justified. Importantly, the authors performed small angle X-ray scattering (SAXS) studies in order to confirm that the PPS state is favored by OM. The solution structures nicely demonstrate that indeed the PPS state is populated by the drug. The manuscript is well written. The scope of the manuscript is interesting for a wider field. Thus, I recommend the manuscript for publication.

However, some minor changes are recommended:

- Page 5: far from the 'PR' site is rather relative, perhaps more precise: on different sides of the lever arm (yellow helix, figure 1e)

- Page 5: The quality of the fit ($\chi^2 = 1.69$) agrees with the conclusion that the OM-S1-PPS structure is the conformation adopted in solution (Fig. 4a). \diamond was the data also fitted with the PR structure? from own experience I know that the saxs profiles of both structures are rather similar, however, they do show differences in the same region (at $\sim 0.12 \text{ \AA}^{-1}$) as the profiles differ in Figure 4b

- The theoretical SAXS curve were... either the theoretical SAXS curve was ... or the theoretical SAXS curves were

- Figure 1: Powerstroke vs Power stroke

- Figure 4:

- better: SAXS data shows, that in solution OM populates...
- OM+ is not defined
- Green and blue (cyan) are a little bit hard to distinguish, perhaps dark blue (as font)
- How is the fit with post-rigor structure
- Axis titles are inconsistent; either $I(q)$ and $q = 4\pi \sin\theta/\lambda$ or $I(s)$ and $s = 4\pi \sin\theta/\lambda$

- Methods: SAXS

This sentence is rather confusing: The PPS APO model for cardiac S1 was built using the motor domain from the APO structure, and building the lever arm helix and ELC from the OM+ structure using the converter position as a reference. why was a PPS APO model built? What is ELC, as mentioned above OM+ not closer defined

- Please comment about number of concentrations that were measured, did the data suggest that all constructs were in monomeric state (were there traces of aggregation or higher oligomeric species? I assume not, since the fit is good at low angles, with exception of the cyan curve(only Mg)).

- It has become best practice to make published SAXS data available. Thus it is strongly recommended that the authors deposit the SAXS data at <https://www.sasbdb.org/> or <http://www.bioisis.net/>

Point-to-point response to the comments of reviewers:

Reviewer #1 (Remarks to the Author):

This is very nice, technically well executed study presenting the structure of the small molecule cardiac myosin activator, omecamtiv mecarbil, bound to cardiac myosin. The structure was found to be locked into the primed state, with OM bound in a pocket (PPS) distinct from that previously observed. This is relevant as the primed structure is the physiologically relevant one for the explaining the kinetic effect of OM on myosin. A detailed analysis of the binding pocket explains why OM is specific for cardiac muscle myosin II. Together, these results significantly advance understanding of the mechanism by which OM activates cardiac myosin, that is by increasing the number of myosin heads bound to actin and prepared to make a power stroke.

We are grateful to Reviewer #1 for its very positive assessment of our work.

My only comment is that the authors should include a discussion of a study on OM recently published in JBC (Sweson et al. Omecamtiv Mecarbil Enhances the Duty Ratio of Human β -Cardiac Myosin Resulting in Increased Calcium Sensitivity and Slowed Force Development in Cardiac Muscle.)

This paper was in fact cited in the draft in several places (ref. 9) regarding the following properties of OM:

- ADP release is unchanged with OM (together with ref. 4);
- Reduced ATPase activity with OM (together refs. 4 and 8);
- Increase of the duty ratio with OM.

We have now added a comment regarding their results for *in vitro* motility in high concentrations of OM (10 μ M). Page 10 includes now the text we have added (green):

“Higher duty ratio also leads to the increase in calcium sensitivity and slow force development in cardiac muscle that have been reported upon experiments with high OM concentrations (10 μ M)⁹. However, given that at clinical doses the effective free fraction of OM in human plasma^{3,15} (<200 nM) is similar to the K_d for heads with MgADP.VO₄ bound and lower than the binding affinity of OM to any other nucleotide state, the findings at high concentrations of OM may not be relevant to its effect on myocardial contractility.”

Reviewer #2 (Remarks to the Author):

Omecamtiv Mecarbil (OM) is in phase 3 clinical trials as a treatment for chronic heart failure. It was discovered more than 5 years ago but despite extensive studies its mode of action remains poorly understood. It binds to the myosin motor domain; it is specific for β -cardiac myosin and it has been shown to increase cardiac output in cardiomyocytes, hence its potential for the treatment of heart failure. There have been three reports over the last year on the effects of OM on the ATPase cycle of human beta cardiac myosin. Yet the mechanism of action remains poorly defined. A crystal structure of human myosin

with OM bound was published in Nat Commun a year ago which placed the OM in a region between the converter domain and the relay helix.

The new report using bovine β -cardiac myosin finds the position of the OM close-by but in a quite distinct conformation (of the motor and the drug) to that previously reported. The motor domain here is in the Pre Power-Stroke conformation favoured by an ADP.Pi containing myosin motor in which the myosin is poised ready to bind actin and undergo a power-stroke. The OM position is inverted compared to the earlier structure. An important feature of the model is that the location is only available at certain points in the ATPase cycle requiring the drug to bind and dissociate during each ATPase cycle.

The crystallography is beautifully presented as expected from this very experienced team. The arguments are logically presented and easy to follow. The correction to the earlier structure is important in this active field of research and the structure may help to understand the mode of action of this potentially significant human drug. However, beautiful the structures the explanations for the mode of action of OM appear to be incomplete or speculative and further information is needed before we can begin to understand how the drug produces all of the effects reported for it. This does not detract from this important work which is an essential step of the way to understanding this drug.

We are equally grateful to Reviewer #2. Although critical of some aspects of our presentation, the reviewer indicates the importance of our results and why the previous structure needed to be corrected.

1. The fact that this work used bovine myosin needs to be more upfront. It is not mentioned in the abstract, title or introduction.

While the use of bovine myosin was stated in the results and methods sections, we agree with the reviewer that the point could be clarified and now we also mention it in the introduction (see pg 4).

“Here, we describe a previously unseen conformation of the bovine cardiac myosin motor in the PPS state that OM stabilizes and binds to with high affinity.”

2. Since the earlier work did use a different source of then the different binding modes of OM could be due to species differences. The conservation of the binding site between human and bovine is well demonstrated in Table 2. Is there any explanation for the different conformations of bound OM in the two structures.

A- Comparison of the human and bovine sequences.

This text has been now added in Methods (see pg 13):

“Note that the bovine and human MYH7 motor domains share 95.9% identity and 98.2% similarity. Out of 830 residues included in the motor domain, the main differences in sequence are found for 13 solvent-exposed residues. Based on this, no difference is to be expected in the structure of human and bovine cardiac myosins and this is supported by the similarity of their kinetic properties as previously reported (Deacon et al.³¹ – see Table 1 ; Liu et al.⁸ – see Table 2) in which human and bovine myosins were directly compared.”

In addition, although the conservation of the ‘PPS’ pocket is shown in Table 2, we did not address this conservation in the text. A new sentence has been added to more clearly indicate this point (see pg 7).

“Additionally, the network of interactions between the drug and the motor in the ‘PPS’ binding site readily explains the specificity of OM action towards cardiac myosin and not other closely related myosins, such as smooth muscle myosin (SMM) and skeletal muscle myosin⁴ (Table 2). The most important packing interactions with OM are accomplished with residues Y164, D168, H492, H666, N711, R712 and I713, which are strictly conserved between bovine and human cardiac myosin (Table 2). Interestingly, the majority of these residues are not conserved in the sequence of other myosin II isoforms (Table 2). In particular, four main residues vary in these sequences: Y164 (Ser in SMM, Phe in Skeletal), H666 (Thr in SMM), N711 (Ser in Skeletal) and I713 (Val in Skeletal). These important differences plainly account for the selectivity of OM for

cardiac muscle myosin II. In contrast, most of the residues found in the 'PR' pocket are conserved among myosin II isoforms (Supplementary Table 3)."

B- Plasticity of the OM molecule when bound to myosin

In addition, note that the residues interacting with OM in the 'PR' pocket observed by Winkelmann et al. are also strictly conserved between bovine and human cardiac myosin (as shown in Supplementary Table 3). Thus, the differences observed in the bound conformations cannot simply be the consequence of the species used.

Moreover, in the OM-PR crystal structure, two independent myosin/OM molecules are found in the asymmetric unit of the crystal. Comparison of these two complexes indicate that the conformation of OM and the binding site are slightly different (see Winkelmann et al. 2015 Nat Comm Fig. 5), highlighting the plasticity of the 'PR' pocket. Thus, the difference in conformation of the OM molecule is linked to its intrinsic plasticity and its adaptation to the site in which it binds. In the OM-PPS crystal structure, the binding site is quite different and the OM molecule adopts a distinct conformation allowing the best fit in this pocket.

Thus the difference in the OM conformation in the two binding sites must be related to the different pockets occupied in the myosin states crystallized. Further, as we showed, the binding affinities of OM to the 'PR' state and the 'PPS' state are quite different reflective of this change in binding site structure. The binding affinity of OM ($0.29 \pm 0.04 \mu\text{M}$, Table 1) to the 'PPS' state is in the range of OM concentrations that is relevant clinically (100 ng/mL to 600 ng/mL, or $0.05 - 0.3 \mu\text{M}$, taking into account the 20% free fraction of OM in plasma) while the binding affinity of OM to the 'PR' state is substantially higher ($5.3 \pm 1.8 \mu\text{M}$) and not achievable clinically before drug intolerance occurs. Thus, we feel that the PPS state binding site is the relevant binding site and that the differences in conformation between the two states, account for the changes in binding affinity.

3. The work provides a structural explanation for the previously reported result that OM stabilizes the M.ADP.Pi complex at the expense of the M.ATP complex a change in the equilibrium constant for the hydrolysis step on myosin.

We also agree that this is an important point. In the introduction, we cite the observation by Liu et al. (Biochemistry 2015):

"The kinetic steps in the motor ATPase cycle that OM influences have been identified. OM shifts the equilibrium of the recovery stroke and the myosin ATP hydrolysis step towards the ADP.P_i bound state, thus increasing the population of heads in the PPS state ready to bind to the actin filament^{3''}.

In the results section, we describe the structural basis for this phenomenon:

"As the 'PPS' OM-binding pocket closes, it buries the drug allowing OM to act as a bridge between the converter domain, the relay helix, and the N-terminus subdomain near the third beta-strand of the transducer. By doing this, OM stabilizes the PPS and thus favours ATP hydrolysis, increasing the number of heads with ADP.P_i at the end of the recovery stroke, ready to commit to binding the actin filament once cardiac systole begins."

4. The argument is presented that OM stabilizes the PPS state by making significant contacts across the converter/ relay helix and N- terminal domains. Thus making more myosin heads available to bind actin from the M.ADP.Pi state – at the expense of the M.ATP pre recovery stroke state. This seems reasonable but this implies that OM also stabilize this M.ADP.Pi state against the following steps that destroy this OM binding pocket (reverse recovery stroke). This is observed for myosin in the absence of actin where Pi release is inhibited. But no such effect is observed in the presence of actin where Pi and is accelerated while the Vmax of the ATPase and the motility is inhibited. **The same forces that act to stabilizing the M.ADP.Pi**

state compared to M.ATP must also resist its loss and would provide the mechanism to inhibit the overall cycling speed reported for OM. However this mechanism would also not be simply reconciled with an increase in duty ratio and higher steady state force and accelerated Pi release. The new structural data does not explain this mechanism.

We argue that the cardiac/myosin structure we provide can explain this mechanism as follows:

One cannot directly compare what would be slow in the absence of actin and what occurs in the presence of actin. The transitions that are slower in the ATPase assays in the absence of actin and presence of OM (stabilization of the pre-powerstroke state and slowing of recovery stroke) are not the same as the transitions that occur on F-actin (Pi and ADP release). Our structure shows that the rate of transitions linked to a swing of the lever arm would open the OM pocket, and thus OM binding would slow the transitions involving a change in the lever arm position. Thus, in the absence of actin OM slows the recovery stroke and stabilizes the PPS state.

However, the presence of actin gives myosin a different path to follow that is favoured kinetically and energetically. The rate of phosphate release must be increased because the state stabilized by OM is closer to the conformation required for myosin binding to actin than the range of states it might explore in the absence of OM as we explain on page 9:

“Prior to the start of contraction, the myosin head can exist in a number of states, some of which cannot bind to the actin filament. OM stabilizes the PPS state of the motor in which the lever arm is primed and ready to bind to actin filament. Thus, there is a larger population of heads with a primed lever arm ready to bind to the actin filament and contribute to force once calcium binds to the troponin-tropomyosin complex to initiate the contractile cycle. Further, by stabilizing this PPS state, OM slows the non-productive turnover of ATP, meaning the release of ADP and P_i, in the absence of a force producing interaction with the actin filament.”

Further, in the presence of actin, however, OM doesn't slow Pi release since this occurs without a significant change in the lever arm swing (Llinas et al., Dev Cell 2015). Again, the Pi release rate increases possibly because the stabilization of the primed state by the drug favours the Pi release transition on F-actin.

The finding that OM can, however, slow the following steps of the powerstroke for which the lever arm swings is consistent with our data (see text below), however the physiological relevance of these data are unclear since the concentration of OM employed in those studies was far greater than what is clinically relevant and the dose response of these effects is unknown. As we discussed above, the affinity of OM to myosin varies significantly based on the conformational state of myosin. In the sarcomere, several myosin heads are influencing each other's kinetics and in physiological concentrations of OM, the drug will not be bound to all heads. Overall, the kinetics of the powerstroke of heads with OM bound will be accelerated by the heads that do not have OM bound.

This is discussed in the current manuscript as follows (see pg 10):

“The mechanism revealed here for OM based on the visualization of its binding site is in agreement with the current model we have proposed in which P_i is released from a state with the lever arm still primed^{10,11}. This mechanism explains why OM increases the rate of Pi release for myosin when OM is bound. The structural data also suggest that OM is released from the motor when the lever arm swings, since the ‘PPS’ pocket opens as the powerstroke proceeds”

[...]

“Recent time-resolved FRET studies¹⁸ under high OM concentrations indicate that after fast Pi release, OM slows the next transition which is associated with the lever arm swing, resulting in an increase of the duty ratio⁹. Our model predicts that OM binding to the ‘PPS’ pocket could indeed slow the transitions that result in opening this site, in particular those that require a lever arm swing”

Finally, as we discuss, cardiac systole probably only requires one or two crossbridge cycles to occur; thus the most important factor to consider is not unloaded sliding velocity (as measured *in vitro*) but ensemble myosin force generated by more engaged crossbridges as shown by Askel, et al (Cell Rep. 2015 May 12;11(6):910-20). This is further supported by the lack of change in the rate of pressure development seen at physiologic exposures in the intact dog model of heart failure (Malik et al, Science 2011).

“In cardiac muscle, the myosin motor cycles at a rate that allows it to undergo only one or two crossbridge cycles during the contractile period of a single heartbeat¹⁷. Further, during systole only about 10-30% of the cardiac myosin heads engage the actin filament to produce force¹⁷. It is clear that a cardiac myosin activator must not significantly slow down several steps of the motor cycle (such as hydrolysis, motor attachment to F-actin and detachment from F-actin) otherwise the ability of the motor to progress through its cycle would likely be impaired. On the other hand, speeding up the motor sufficiently to allow it to undergo more crossbridge cycles during a single heartbeat is probably not feasible given that one would need to accelerate the cycle substantially to do so. OM appears to take advantage of another means of increasing contractile force; it increases the number of myosin heads that engage the actin filament during the cardiac contraction.”

5. The authors argue that their most recent model of a cycle Pi release before a power-stroke/lever arm swing is compatible with the proposed mode of action of OM. This is compatible with OM inhibiting the power stroke but there is no explanation of how the model can explain the actin induced acceleration of the Pi release step. The model of Rohde et al appears equally probable.

OM can increase the rate of Pi release in two ways: it increases the population of heads with a primed state and ADP.Pi trapped ready to bind to actin; and it also stabilizes the states of the motor with a lever arm primed thus favouring the transitions that occur on F-actin without any swing of the lever arm. It is thus likely that OM lowers the energy of the Pi release state on F-actin and thus increases the rate of Pi release. The Rohde et al. model as shown in Fig 7B is very similar to the model proposed in Llinas et al Dev Cell 2015. However, the Rohde et al. model also proposes that in the absence of OM another pathway would be favoured: the release of Pi would occur after the powerstroke swing rather than prior to the powerstroke (Fig 7A of their paper). In the presence of OM, Pi release would occur from a different structural state in which the lever arm would be up, which is essentially the same model as the one we propose in the manuscript based on the Llinas et al. paper.

A simpler, and we believe more likely, proposal is that the same transition occurs in both cases, however the kinetics are faster in the presence of OM because it stabilizes the states that facilitate this transition occurring on F-actin.

It is important to note that the experiments in Rohde et al. can characterize with high temporal resolution the swing of the lever arm but they are not demonstrating that two pathways occur since the probes could also monitor two successive transitions. There is a lack of spatial information of what occurs to the heads during these time transitions since such FRET data cannot directly give access to the amount of structural changes sensed by the probes. In addition, the data presented for cardiac myosin in the Rohde et al. paper does not indicate that Pi release would be slower than the lever arm swing in the absence of OM : the rate of Pi release and the fast rate of the powerstroke are similar (Rohde et al paper Figs 3, 4).

In conclusion, we believe that the model we propose, built from a Pi release transition that would occur with little lever arm swing (Llinas et al.) is sufficient to interpret our data.

6. At the moment the model appears to require simultaneous release of OM and ADP at completion of the lever arm swing. Is this possible? If the lever arm swing allows both events then OM released like ADP release will be load dependent. Inhibiting the lever arm swing would give higher force in the steady-state but also slow the cycle and ADP release under zero load.

After Pi release, the swing of the lever arm is not coupled with ADP release; this occurs later at the end of the powerstroke. It is likely that the first swing of the lever arm would open the OM pocket and thus it is likely that OM diffuses away at this step. The ADP release step itself has been demonstrated to be non-affected by the presence of OM (refs 4, 8, 9 and 18), which fits with this mechanism. We believe higher force is generated based on an increase in ensemble force for the reasons outlined above.

7. It seems that a more holistic look at the cycle and where OM might affect the cycle with and without load will be needed before we understand the role of OM

We agree with the reviewer for this important point. This is why in discussion we prefer to focus on the effect in the sarcomere (as an ensemble of myosin heads) and at physiological concentrations of the drug. This is what is relevant for the use of the drug therapeutically.

8. If the PPS, M.ADP.Pi state is stabilized it might be predicted stabilize the “J-motif” or “off-state” of the thick filament making it harder to activate the thick filament –leading to lower force. REF Hooijman P, Stewart MA, Cooke R (2011) Biophys J 100:1969-1976.

This is a possibility. Unfortunately the current resolution for the J-motif is too low to state whether the OM binding site would be conserved in these heads. Note that the blocked and free heads do not adopt the same conformation for their lever arm. It is possible, in fact, that these conformations are not compatible with OM binding as the position of the lever arm in the two heads required for stabilizing the off-state may not be compatible with OM binding, which requires a particular position of the lever arm and the motor domain. Further experiments will be required to investigate this point.

9. This M.ADP.Pi state is also proposed to be the form of myosin interacting via the myosin mesa with MyBP-C. The OM could interfere alter the MyBP-C interaction.

MyBP-C is proposed to bind to cardiac myosin II via its S2 region (EM structure: Al-Khayat et al., 2013) and a report has also proposed that the mesa surface of its motor domain might be involved in its interaction in relaxed conditions (Spudich, 2015). This interaction occurs in the super-relaxed (SRX) state in which the two heads of myosins form a so called “J-motif” composed of a blocked and a free-head (Alamo et al., 2016) that seems conserved amongst myosin 2 in multiple organisms and in myosins from several filaments : tarantula striated fiber (Alamo et al., 2008; Alamo et al., 2016); Schistosoma mansoni worm smooth fiber (Sulbaran et al., 2013); chicken gizzard smooth muscle (Wendt et al., 2001); human, mouse and zebrafish cardiac fibers (Al-Khayat et al., 2013; Zohgbi et al., 2008; Gonzalez-Sola et al., 2014); scorpion striated fibers (Pinto et al., 2012), Limulus (Zhao et al., 2009) and scallop striated filaments (Woodhead et al., 2013). However, there is a still reservation about the correspondence between the interhead motif and the SRX state. In addition, the resolution described for these structures is insufficient to conclude about how strictly conserved the motif is and whether the heads adopt a canonical PPS or not. It is thus also impossible to get from these structures insights about the position of the converter and its exact position compared to the motor domain (28 Å for cardiac filament [code EMDB: EMD-2240] and 20 Å for top-resolution structure of tarantula filament [code EMDB: EMD-1950/ code PDB: 3JBH]).

10. If OM does stabilize the M.ADP.Pi state at the expense of M.ATP the drug may be useful in correcting the problem with HCM mutations in which the mutation inhibits the ATP hydrolysis step thus favoring M.ATP over M.ADP.Pi. i.e. the classic R453C mutation associated with severe HCM.

We believe, like the reviewer, that indeed some HCM mutations would benefit from the use of OM. Although the effects of the addition of a small molecule will require more investigations, we agree with this reviewer that important future studies will require characterizing the impairment lead from disease mutations as well as matching these mutations with different activators/inhibitors of myosin. Novel treatments are thus now feasible for severe cardiac diseases.

Reviewer #3 (Remarks to the Author):

Planelles-Herrero et al. describe the crystal structures of cardiac myosin in the pre-powerstroke state (PSS), both with (2.45 Å) and without (3.0 Å) the selective activator Omecamtiv mecarbil (OM) bound. This allows them to derive mechanism of action of OM. The binding of OM does not significantly alter the structure of the motor domain. Instead it binds to an allosteric site so that the lever arm rests in a primed position increasing the population of heads in the PSS state ready to bind to the actin filament. This is especially interesting as previously a different binding site of OM was described by Winkelmann et al 2015 (in Nature Communication). However, this structure did not provide a rationale for the selectivity of OM for cardiac myosin, as compared to other muscle myosins. In contrast, the new binding site not only accounts for the selectivity of the drug for cardiac myosin but it also provides a strong rationale for the mechanism of cardiac myosin activation which is in strong agreement with previous kinetic studies. The data seems sound and the conclusions are well justified. Importantly, the authors performed small angle X-ray scattering (SAXS) studies in order to confirm that the PSS state is favored by OM. The solution structures nicely demonstrate that indeed the PSS state is populated by the drug. The manuscript is well written. The scope of the manuscript is interesting for a wider field. Thus, I recommend the manuscript for publication.

We are very grateful to reviewer #3 for the positive comments and her/his enthusiasm regarding our manuscript.

However, some minor changes are recommended:

- Page 5: far from the 'PR' site is rather relative, perhaps more precise: on different sides of the lever arm (yellow helix, figure 1e)

We agree that the sentence "far from the 'PR' site" is rather vague. We have changed the sentence as follows (see pg. 5).

"In the OM-S1-PPS structure, OM binds in a previously unseen pocket of the motor, which we call the 'PPS' allosteric site. This 'PPS' pocket is not only separated by more than $\sim 18\text{\AA}$ from the previously described site in the PR state¹² ('PR' pocket), but also its environment is completely different (Fig. 1e)".

- Page 5: The quality of the fit ($\chi^2 = 1.69$) agrees with the conclusion that the OM-S1-PPS structure is the conformation adopted in solution (Fig. 4a). Was the data also fitted with the PR structure? from own experience I know that the saxs profiles of both structures are rather similar, however, they do show differences in the same region (at $\sim 0.12\text{\AA}^{-1}$) as the profiles differ in Figure 4b

The reviewer is right, when the PR structure is fitted into the MgADP.VO₄+OM data, the major differences are observed at $\sim 0.1-0.16\text{\AA}^{-1}$. However the differences are quite important, leading to a decrease in the fit quality ($\chi^2 = 117.61$).

Note that to date there is no available PR structure with the light chain (the cardiac PR structures in the PDB all have a linker and a C-terminal GFP after the converter). In order to perform the fit, we created a chimeric structure using our lever arm+light chain structure (mutating the residues to match the bovine sequence), using the converter of the PR structure as a reference to position the lever arm.

- The theoretical SAXS curve were... either the theoretical SAXS curve was ... or the theoretical SAXS curves were

We have changed this sentence (pg. 13):

"The theoretical SAXS curves were calculated with CRYSQL³² and compared based on the quality of their fits against the different experimental curves."

- Figure 1: Powerstroke vs Power stroke

We have modified Figure 1 and Figure 5 and changed “Power stroke” to “Powerstroke”.

- Figure 4:

- better: SAXS data shows, that in solution OM populates...
- OM+ is not defined
- Green and blue (cyan) are a little bit hard to distinguish, perhaps dark blue (as font)
- How is the fit with post-rigor structure
- Axis titles are inconsistent; either $I(q)$ and $q = 4\pi\sin\Theta/\lambda$ or $I(s)$ and $s = 4\pi\sin\Theta/\lambda$

We thank the reviewer for pointing this out. We have modified figure 4 and its legend to satisfy all these issues.

- Methods: SAXS

This sentence is rather confusing: The PPS APO model for cardiac S1 was built using the motor domain from the APO structure, and building the lever arm helix and ELC from the OM+ structure using the converter position as a reference. why was a PPS APO model built? What is ELC, as mentioned above OM+ not closer defined

We agree that the sentence was confusing. We have modified page 13 (Methods) as follows:

“As the APO-PPS-MD structure lacks the lever arm and the light chain, a chimeric model was constructed using the motor domain from the APO-PPS-MD structure, and building the lever arm helix and the light chain from the OM-S1-PPS structure using the converter position as a reference to position the lever arm.”

- Please comment about number of concentrations that were measured, did the data suggest that all constructs were in monomeric state (were there traces of aggregation or higher oligomeric species? I assume not, since the fit is good at low angles, with exception of the cyan curve(only Mg)).

We did measure different concentrations for the SAXS experiments (~2 and ~5 mg/ml). However, for each condition, both curves were identical although the signal/noise ratio is better for the data measured at 5 mg/ml. Thus, we only used the 5 mg/ml curve of each condition for further analysis. We have modified the methods (page 13) to satisfy reviewers' concerns.

- It has become best practice to make published SAXS data available. Thus it is strongly recommended that the authors deposit the SAXS data at <https://www.sasbdb.org/> or <http://www.bioisis.net/>

We thank this reviewer for pointing out the SAXS data repositories. Indeed, we were not aware of the existence of these databases. As we fully agree with its philosophy, we have deposit the MgADP.VO₄+OM (**SAS299**) and the MgADP+OM (**SAS300**) data.

We have added a sentence in the Data Availability section (pg. 13):

“The atomic coordinates and structure factors have been deposited in the Protein Data Bank, www.pdb.org, with accession numbers **5N69 (OM-S1-PPS)** and **5N6A (APO-MD-PPS)**. The SAXS data with OM has been deposited in the Small Angle Scattering Biological Data Bank (SASBDB), www.sasbdb.org, with accession numbers **SAS299 (MgADP.VO₄+OM)** and **SAS300 (MgADP+OM)**.”

REVIEWERS' COMMENTS:

Reviewer #2 (Remarks to the Author):

This is a resubmitted manuscript. In my initial review I stated that "The crystallography is beautifully presented as expected from this very experienced team. The arguments are logically presented and easy to follow. The correction to the earlier structure is important in this active field of research and the structure may help to understand the mode of action of this potentially significant human drug. However, beautiful the structures the explanations for the mode of action of OM appear to be incomplete or speculative and further information is needed before we can begin to understand how the drug produces all of the effects reported for it. This does not detract from this important work which is an essential step of the way to understanding this drug."

The authors have made a detailed response to my original critique of the mechanism proposed and made some adjustments to the text. While there remain questions over the detailed mechanism I believe this is an important paper that should be published